# Influence of Zinc on *Histoplasma capsulatum* Planktonic and Biofilm Cells

**DOI:** 10.3390/jof10050361

**Published:** 2024-05-20

**Authors:** Ana Carolina Moreira da Silva Pires, Angélica Romão Carvalho, Carolina Orlando Vaso, Maria José Soares Mendes-Giannini, Junya de Lacorte Singulani, Ana Marisa Fusco-Almeida

**Affiliations:** 1Department of Clinical Analysis, School of Pharmaceutical Sciences, São Paulo State University (UNESP), Araraquara 14800-903, SP, Brazil; a.romao.carvalho@gmail.com (A.R.C.); carolovaso@hotmail.com (C.O.V.); maria.giannini@unesp.br (M.J.S.M.-G.); junyadelacorte@yahoo.com.br (J.d.L.S.); 2Department of Microbiology, Institute of Biological Sciences, Universidade Federal de Minas Gerais, Belo Horizonte 31270-901, MG, Brazil

**Keywords:** *Histoplasma capsulatum*, histoplasmosis, biofilm, metallic ions, TPEN, zinc

## Abstract

*Histoplasma capsulatum* causes a fungal respiratory disease. Some studies suggest that the fungus requires zinc to consolidate the infection. This study aimed to investigate the influence of zinc and the metal chelator TPEN on the growth of *Histoplasma* in planktonic and biofilm forms. The results showed that zinc increased the metabolic activity, cell density, and cell viability of planktonic growth. Similarly, there was an increase in biofilm metabolic activity but no increase in biomass or extracellular matrix production. N′-N,N,N,N–tetrakis–2-pyridylmethylethane–1,2 diamine (TPEN) dramatically reduced the same parameters in the planktonic form and resulted in a decrease in metabolic activity, biomass, and extracellular matrix production for the biofilm form. Therefore, the unprecedented observations in this study highlight the importance of zinc ions for the growth, development, and proliferation of *H. capsulatum* cells and provide new insights into the role of metal ions for biofilm formation in the dimorphic fungus *Histoplasma*, which could be a potential therapeutic strategy.

## 1. Introduction

Histoplasmosis is a fungal respiratory disease with a high incidence and mainly affects immunocompromised individuals [1,2,3]. The pathogen is the dimorphic fungus *Histoplasma capsulatum*, which grows in soils rich in organic matter, especially in guanos from birds and bats [4,5,6]. The World Health Organization (WHO) recently determined a ranking for prioritizing fungal pathogens using criteria such as the number of deaths, annual incidence, global distributions, and complications. Fungi such as *Candida* spp. and *Histoplasma* are present in high-priority groups [7].

*H. capsulatum* pathogenesis occurs due to the host’s inhalation of microconidia or hyphae fragments, in which these structures will be deposited in the bronchi and pulmonary alveoli [8]. The change in environment temperature (28 °C) to the mammalian body temperature (37 °C) induces the morphogenesis from the fungal mycelial form to the yeast form [2]. The host has defense mechanisms related to the production of reactive oxygen species (ROS) by macrophages and the limiting of essential elements and micronutrients by the host’s immune system during the invasion by pathogenic microorganisms. These mechanisms are called “nutritional immunity”; several virulence factors protect the fungus from surviving in phagocytes [9,10]. *H. capsulatum* has several virulence factors, such as thermal dimorphism, the expression of surface proteins, and the formation of biofilms that contribute to its adhesion, colonization, and survival in the host immune system [11,12,13,14].

Biofilms are classified as communities of microorganisms that develop on living or inert surfaces and are surrounded by an extracellular matrix containing polysaccharides and other elements [15,16,17]. This formation provides a physical barrier, impairs antimicrobials’ penetration, and is associated with greater resistance to host immune mechanisms [12,18,19,20]. Biofilm formation occurs in a sequence of processes that will overlap during its development. Despite using sequenced and labeled stages of biofilm formation and development, the processes that occur under native conditions are much more complex, dynamic, and diverse [21]. Many authors have demonstrated that the addition of metal ions or the removal of these ions disrupts the formation of biofilm [22,23,24]. Studies have shown that metal ions alter the structures of *Candida tropicalis* and *C. albicans* biofilms and increase the transition between yeast form and hypha cells. This increases invasiveness and alters virulence [25].

The metal ion homeostasis in infection has gained visibility due to recent studies [22,23,24,25,26]. The host sequesters metal ions to prevent the pathogen from obtaining essential ions to replicate inside the cells [27,28]. Metal ions are micronutrients that can play several fundamental cellular roles for host defense and attacks by microorganisms. Among the metallic ions, we highlight zinc [24]. Zinc is an indispensable ion for many cellular activities, such as gene expression, enzymatic activity, and cell signaling, in addition to being related to proliferation, deoxyribonucleic acid (DNA) and ribonucleic acid (RNA) synthesis, and apoptosis [29]. Zinc depletion conditions are known to reduce fungal growth, and evidence shows that host cells sequester intracellular zinc to make access to this micronutrient difficult [30]. Some fungi use transporters to capture the free zinc in an extracellular environment in zinc deprivation or stress conditions. These zinc-binding enzymes are distributed in all cellular components, mainly mitochondria, and the cytoplasm, which characterizes the connection of zinc with the cell metabolism of these microorganisms [31]. 

Few studies related to micronutrients in *H. capsulatum* strains have been described. However, it was demonstrated that the fungus needs zinc to succeed during infection in murine models [32,33]. New studies showed that *Histoplasma* has zinc transport regulators that are homologous to those found in the fungus *Saccharomyces cerevisiae* and other fungi such as *Candida albicans* and *Aspergillus fumigatus* [31], and it also needs the ion as a cofactor of the enzyme superoxide dismutase (SOD) to respond to ROS produced by macrophages as a line of defense [10]. Recently, eight genes involved in zinc homeostasis in *Histoplasma* were identified in an in silico analysis. The authors demonstrated the influence of zinc starvation in several metabolic pathways, mainly in the metabolism of glucose, biosynthesis of structural carbohydrates, biosynthesis of pyridoxine, and regulation of oxidative stress [34].

Therefore, due to the lack of studies on metal ions in *H. capsulatum*, this work is an unprecedented study that aims to evaluate the influence of zinc deficiency and abundance on the growth of *H. capsulatum* strains in planktonic and biofilm forms to monitor the effects of zinc.

## 2. Materials and Methods

### 2.1. Strains and Growth Conditions

The assays used two strains of *H. capsulatum* (ATCC 26029 (G186 A) and EH-315 (BAC1)). The strain ATCC 26029 (G186A), originally from Panama, is a clinical strain, and EH-315 (BAC_1_) is a wild strain in the World Federation for Culture Collection database under the number LIH-UNAM WDCM817. The *H. capsulatum* strains were maintained for 96 h at 37 °C on Brain and Heart Infusion (BHI) agar (BD Difco™, Wokingham, Berkshire, UK) supplemented with 0.1% L-cysteine (Synth, Diadema, Sao Paulo, Brazil) and 1% glucose (Sigma-Aldrich, Milano, Italy) [35]. 

The strains ATCC (G186A) and EH 315 were removed from BHI agar and subcultured at 37 °C for 96 h in Ham’s Nutrient Mixture medium (HAM-F12) (Gibco^®^, Thermo Fisher Scientific, Waltham, MA, USA) supplemented with 1.8% glucose (Synth, Diadema, Sao Paulo, Brazil), 0.1% glutamic acid (Synth, Diadema, Sao Paulo, Brazil), 0.6% buffer HEPES (Sigma-Aldrich, Milano, Italy), and 0.008% L-cysteine (Synth, Diadema, Sao Paulo, Brazil), with an agitation of 150 rpm [36]. For the zinc (Zn) deprivation experiments, the yeasts were grown in HAM-F12 medium supplemented with 10 µM of a solution of N′-N,N,N,N–tetrakis–2-pyridylmethylethane–1,2 diamine (TPEN) chelator (Sigma-Aldrich, Milano, Italy). For excess zinc conditions, zinc sulfate heptahydrate (ZnSO_4_·H_2_O) was added in a concentration of 20 µM in HAM-F12. The study groups were defined as HAM-F12 (as control group), HAM-F12 + 20 µM Zinc, and HAM-F12 + 10 µM TPEN. 

### 2.2. TPEN Chelator Inhibition Profile 

Assays of the inhibition profile of the chelator were performed according to Vaso et al. [37] with modifications. The TPEN chelator was dissolved separately in 50 mL of HAM-F12 medium to reach a concentration of 2.5 µM. Then, a microdilution of this chelator was performed in 96-well plates, with concentrations ranging from 0.312 µM to 80 µM. The initial inoculum was carried out in sodium phosphate butter (PBS) at a concentration of 5 × 10^6^ cells/mL, with subsequent dilution of 1:10 in HAM-F12 medium. One hundred microliters of the inoculum were added to 96-well plates containing 100 µL of the chelators in serial dilutions. Controls were performed for the medium’s sterility and fungal growth. The plates were incubated at 37 °C for 24 h, and viability was determined using 20 µL of 0.02% resazurin (Sigma-Aldrich, Milano, Italy). Readings were performed on the Epoch Bioteck^®^ Spectrophotometer at a wavelength of 570 nm/600 nm. 

### 2.3. Planktonic Assay

After the growth of the fungus, the strains were centrifuged at 5000 rpm for 10 min. The supernatant containing the culture medium was removed, and the fungal mass was resuspended with PBS for washing and counting the cells in a Neubauer chamber with Trypan blue (Gibco^®^, Thermo Fisher Scientific, Waltham, MA, USA) at a ratio of 1:1. The experiments were carried out with a viability of at least 90%. Then, the cells were resuspended in different media: HAM-F12 (control), HAM-F12 + 20 µM zinc, and HAM-F12 + 10 µM TPEN. The initial concentration used was 5 × 10⁶ cells/mL, and these 96-well plates with media and the inoculum were incubated under agitation at 150 rpm at 37 °C for 144 h. The summary flowchart is shown in Figure A1 (Appendix A).

#### 2.3.1. Optical Density (OD)

Optical density at a wavelength of 520 nm was measured in three wells at 6, 12, 24, 48, 72, 96, 120, and 144 h using a spectrophotometer (Biotek—Epoch 2™, Agilent, Santa Clara, CA, USA) [14]. 

#### 2.3.2. Measurement of Metabolic Activity

The mitochondrial activity of the biofilm and the planktonic growth of the fungus was measured under all conditions using the XTT reduction assay (2,3-bis-(2-methoxy-4-nitro-5-sulphenyl)-5-(phenylamine carbonyl)]-2H—tetrazolium hydroxide) (XTT). The times of 0, 6, 12, 24, 48, 72, 96, 120, and 144 h of incubation of planktonic growth were evaluated. The XTT solution was added without removing the medium in the planktonic form. Fifty microliters of the XTT solution and 4 µL of menadione solution were added to the wells of the plate and subsequently incubated at 37 °C for 3 h. Subsequently, the optical density of the wells was read in a spectrophotometer (Biotek Epoch 2™) at 490 nm [12,19]. 

### 2.4. Biofilm Assay

With some modifications, biofilms were formed as established by Pitangui et al. [12]. The inoculum was resuspended in PBS, and the viability of the fungal cells was checked. The initial inoculum was prepared in PBS at a concentration of 5 × 10^6^ cells/mL. Two hundred microliters of the inoculum prepared under each condition (HAM-F12 (control), HAM-F12 + 20 µM zinc, and HAM-F12 + 10 µM TPEN) was placed in each well of a 96-well plate. The plates were incubated for 144 h under aerobic conditions at 37 °C without agitation for biofilm formation. The summary flowchart is shown in Figure A2 (Appendix A).

#### 2.4.1. Measurement of Metabolic Activity

In the biofilm, the times of 6 h, 12 h, 24 h, 48 h, 96 h, 120 h, and 144 h were evaluated using the XTT reduction assay previously described Section 2.3.2. The supernatants were carefully removed, and the wells were washed with sterile PBS to remove medium or non-adherent cells. 

#### 2.4.2. Measurement of Biofilm Biomass

Following the exact evaluation times of metabolic activity, the amount of biomass produced by the biofilms under the estimated conditions was evaluated. In each incubation period determined for the biofilm, the supernatants were removed from the wells, and 200 µL of methanol (Sigma-Aldrich) was added for 15 min. The methanol supernatant was aspirated, and the plates were dried at room temperature in laminar flow. Then, 200 µL of a 0.1% crystal violet solution (Dinamica) was added to each well for 20 min. After this period, the dye was removed and discarded, and the biofilms were washed with sterile distilled water and fixed with 200 µL of 33% acetic acid. Then, the plates were read in a Biotek Epoch 2™ spectrophotometer at 570 nm [35].

#### 2.4.3. Measurement of Biofilm Extracellular Matrix

The extracellular matrix of biofilms was quantified by staining with safranin. The supernatants from the wells were discarded, and 50 µL of 1% safranin solution was added to the wells for 5 min. The dye was removed, and the plates were washed with sterile distilled water until the excess dye was removed. Two hundred microliters of sterile PBS (phosphate buffer) was placed in wells. The plate was read on a Bioteck Epoch 2™ spectrophotometer at a wavelength of 492 nm [38]. 

#### 2.4.4. Scanning Electron Microscopy (SEM)

For scanning electron microscopy (SEM), following the previously determined conditions, biofilms were formed in 24-well plates. Biofilms formed in 24 h (initial) and formed in 144 h (mature) were evaluated under all conditions. Samples were processed as Morris et al. [39] described, with some modifications for adaptation to the conditions. Therefore, the supernatant was removed from each plate’s wells and washed with PBS three times to remove non-adherent cells. The concentrations were fixed with 250 µL of 2.5% glutaraldehyde solution (Sigma-Aldrich) and 250 µL of 4% paraformaldehyde solution (Sigma-Aldrich) for 24 h at 4 °C. After fixation, the wells were washed thrice using sterile 0.85% saline water. Then, the dehydration step was carried out, adding drying at room temperature. The bottoms of the wells were cut from the plates with a scalpel, mounted in silver aluminum cylinders, and placed in high-vacuum evaporators for coating with gold (20 Au). The samples were analyzed with JEOL Scanning Electron Microscope model JSM-6610LV from the Scanning Electron Microscopy Laboratory of the School of Dentistry, FOAR, Araraquara, SP, Brazil. 

#### 2.4.5. Confocal Fluorescence Microscopy (CLSM)

Biofilms were also analyzed for density and viability by confocal fluorescence microscopy. Biofilms were formed in 24-well plates following the same conditions previously reported (control, Zinc, TPEN) and at the exact times used for initial and mature biofilms subjected to SEM. After biofilm formation, the supernatant was removed, and the biofilms were fixed with 4% paraformaldehyde for 24 h at 4 °C. Then, the paraformaldehyde was removed, the wells were washed with PBS, and 250 µL of Calcofluor white reagent (Sigma-Aldrich) was added at a 50 µg/mL concentration for 40 min. After the staining time, the wells were washed with PBS, and the samples were analyzed at a wavelength of 380 nm using a confocal fluorescence microscope (Carl Zeiss LSM 800) of the School of Dentistry, UNESP, Araraquara, SP, Brazil; analyses were performed using Zen Blue 3.2 software (Carl Zeiss, Jena, Germany).

### 2.5. Congo Red Assay

The experiments were performed according to Ram and Klis (2006) [40] with minor changes. A susceptibility test was performed using Congo Red (CR) dye (Sigma-Aldrich, Saint Louis, MO, USA) at 100 to 2500 µM concentrations. After determining which concentrations the fungus was sensitive to, the dye was added to the supplemented BHI at 34.8 mg/L (100 µM) and 52.25 mg/L (150 µM) and was plated on Petri dishes. The strains were cultivated in their respective HAM-F12 media in planktonic growth and biofilm form at 24 and 144 h. After these periods, the cultures of each strain were centrifuged and resuspended at concentrations of 5 × 10^5^ cells/mL and 5 × 10^6^ cells/mL, and 5 µL of each concentration, treated and non-treated with CR, was plated onto BHI.

### 2.6. Statistical Analysis

The statistical analysis of the results was performed using the GraphPad Prisma 5 Program (La Jolla, CA, USA). The results were analyzed by analysis of variance (ANOVA) with Bonferroni posttest. The value *p* < 0.05 was considered statistically significant. Two or three independent experiments were performed with three replicates for all tests. 

## 3. Results

### 3.1. Fungal Growth Inhibition Profile with TPEN

First, we determined the inhibition profile of different concentrations of TPEN on each *H. capsulatum* strain. The TPEN chelator was able to reduce the viability of both strains at concentrations ≥10 µM compared to the controlled growth (*p* < 0.001) (Figure A3). Thus, 10 µM of TPEN chelator was the lowest concentration that inhibited fungal viability and was selected for further experiments. 

### 3.2. Influence of Zinc and TPEN on Histoplasma Growth in Planktonic Form 

#### Characterization of Planktonic Growth with Metabolic Activity and Optical Density

The results of metabolic activity and optical density of planktonic growth of *Histoplasma* strains are shown in Figure 1. The metabolic activity of G186A in a medium with zinc was higher than that in HAM-F12 (control) from 24 to 144 h (*p* < 0.001). On the other hand, the strain showed reduced metabolic activity compared to the control from 48 to 144 h when treated with TPEN (*p* < 0.001) (Figure 1A). For EH-315, adding zinc to the medium did not influence its metabolic activity; the strain showed a statistical decrease in metabolic activity compared to that in HAM-F12 medium (control) at times 120 and 144 h (*p* < 0.001). In the medium with TPEN, the metabolic activity of the strain remained low after 24 h; however, a significant statistical difference was observed at 120 and 144 h (*p* < 0.001) compared with the control condition (Figure 1C). According to the optical density results (Figure 1B,D), both strains showed an increase in cell density in the medium with zinc. For strain G186A, the cell density was more notable in the medium with zinc at times from 48 (*p* < 0.01) to 144 h (*p* < 0.001). For strain EH-315, there was an increase in density in the medium with zinc at 120 and 144 h (*p* < 0.001). With TPEN, both strains had a decrease in their cell density. For G186A, the reduction was significant from 24 (*p* < 0.01) to 144 h (*p* < 0.001), and for EH-315, the reduction was significant from 48 to 144 h (*p* < 0.001).

### 3.3. Influence of Zinc and TPEN on Histoplasma Growth in Biofilm Form

#### 3.3.1. Characterization of Biofilm with Metabolic Activity, Quantification of Biomass, and Quantification of Extracellular Matrix 

The results of the characterization of biofilm are shown in Figure 2. The G186 A strain showed higher metabolic activity in the medium with zinc from 24 to 144 h (*p* < 0.001) compared to the control condition. With TPEN added in the medium, the metabolic activity of the strain was lower from 12 to 144 h. The statistical difference was from 72 (*p* < 0.01) to 144 h (*p* < 0.001) (Figure 2A). For EH-315 with the added zinc in the medium, the metabolic activity remained similar to the control (HAM-F12), and only at 144 h (*p* < 0.01) was there an increase in the metabolic activity of the fungus in the control. Nonetheless, the medium with TPEN influenced the activity of the strain, which was lower from 72 (*p* < 0.01) to 144 h (*p* < 0.001) (Figure 2D). Zinc and the control (HAM-F12) showed similar biomass production for both G186A and EH-315 strains. The presence of the chelator reduced the biomass production of the G186 A strain at 48 (*p* < 0.001) and 120 h (*p* < 0.01) and that of the EH-315 strain at 6 (*p* < 0.01) and 12 h (*p* < 0.001) (Figure 2B,E). The extracellular matrix (MEC) quantification showed that the addition of zinc in the medium influenced the MEC production of G186A (Figure 2C) but not EH-315 (Figure 2F). G186A presented an increase in MEC production at 72 (*p* < 0.01), 96 (*p* < 0.001), and 120 h (*p* < 0.05) in the medium with zinc. In the medium with TPEN, the MEC production was lower than the control at 12 (*p* < 0.05) and 24 h (*p* < 0.01) for the G186 A strain and at 6 (*p* < 0.05) and 12 h (*p* < 0.01) for EH-315.

#### 3.3.2. Scanning Electron Microscopy (SEM) and Confocal Microscopy (CSLM) of Histoplasma Biofilm

The morphology and architecture of biofilms of G186A and EH-315 were observed using SEM and CLSM, as shown in Table 1 and Table 2, respectively. 

In the HAM-F12 medium, for the G186A strain at the 24 h time point, cells with intact contours and oval morphology were observed (yellow arrow), forming a biofilm with a thickness of around 200 µM (Table 1A,D,G). In the HAM-F12 medium with added zinc, cells with intact and oval contours were also present (yellow arrow), and the biofilm had a thickness of 200 µM. However, some cells showed the beginning of filamentation (green arrow) (Table 1B,E,H). In the TPEN chelator medium, cells also exhibited oval morphology but were in reduced quantification (Table 1C,F,I). After 144 h, microscopy was conducted again to observe biofilm maturation. In both the control medium (Table 1J,M,P) and the condition of HAM-F12 medium with added zinc (Table 1K,N,Q), a massive cell cluster was observed at the 144 h time point, with the presence of oval cells and hyphae, along with a thickness of 200 µM. However, in the medium containing the chelator (Table 1L,O,R), cells were morphologically affected after 144 h of cultivation and exhibited reduced growth. 

SEM and CLSM images of the EH-315 strain are displayed in Table 2. In both control conditions (HAM-F12) (Table 2A,D,G) and HAM-F12 with added zinc (Table 2B,E,H), observation reveals the presence of cell clusters, indicating the formation of an initial small biofilm at 24 h. This biofilm consists of oval cells (yellow arrow) and some hyphae (green arrow), with a thickness of up to 200 µM. Notably, when zinc is introduced to the HAM-F12 medium, the elongation of yeast cells becomes more pronounced. When HAM-12 is supplemented with TPEN, at 24 h (Table 2C,F,I), a shift is observed with a prevalence of more dispersed oval cells and a lower occurrence of hyphae. Over a growth period of 144 h, the EH-315 strain demonstrates an increased biofilm formation in both the control (Table 2J,M,P) and the zinc-enriched medium (Table 2K,N,Q). These biofilms maintain oval cells and possess a thickness of 200 µM. Furthermore, the medium containing TPEN (Table 2L,O,R) experiences a reduction in cell count and the presence of both hyphae and yeasts. This phenomenon notably differs from the 24 h conditions across all scenarios and conditions. 

#### 3.3.3. Susceptibility to Congo Red (CR) Dye to Determine the Interference of TPEN and Zinc in the Cell Wall

The aim of the Congo Red assay was to evaluate the morphology and viability of colonies derived from yeast spots cultivated in HAM-F12, HAM-F12 with zinc, and HAM-F12 with TPEN. 

The results of the CR assay showed that in the 24 h planktonic growth of the G186A strain (Figure 3), the yeasts in the HAM-F12 medium (control) demonstrated successful growth in pure BHI and BHI with 100 and 150 µM CR. However, yeast in the HAM-F12 medium with zinc and TPEN demonstrated difficulty in growing in the medium with CR at both 100 and 150 µM. Yeast cultivated for 144 h in all HAM-F12 media showed the ability to form spots in BHI and BHI with 100 and 150 µM CR. In biofilm growth of strain G186A, cells were collected and resuspended at the same concentrations as in planktonic culture. Yeast from HAM-F12 medium with zinc and the control formed spots in pure BHI medium and in BHI medium with 100 and 150 µM CR after 24 h and 144 h. However, yeast grown in HAM-F12 medium with the chelator only formed spots in BHI medium and BHI medium with 100 and 150 µM CR when collected after 144 h of growth (Figure 4).

Figure 5 and Figure 6 offer visual depictions of colonies formed by the EH-315 strain after being cultured in planktonic and biofilm growth for 24 h and 144 h. Yeasts from the EH-315 strain cultivated in HAM-F12 medium under both planktonic and biofilm conditions and across the 24 h and 144 h time points exhibited the capability to generate spots. Notably, yeast spots originating from the 24 h planktonic growth were relatively smaller when exposed to concentrations of 100 and 150 µM CR, at a density of 5 × 10^5^ cells/mL. Yeasts grown in HAM-F12 with zinc during the 24 h planktonic growth did not form spots in pure BHI or BHI with 100 and 150 µM CR. However, these yeasts eventually formed small spots after 144 h of cultivation (Figure 5). 

Lastly, yeast sourced from HAM-F12 medium with TPEN, after 24 h and 144 h planktonic growth and 24 h biofilm growth, was incapable of forming spots in BHI media with concentrations of 100 and 150 µM CR—only achieving this in pure BHI. The sole instance of growth observed for yeasts cultivated in the BHI medium occurred after 144 h of biofilm cultivation, with spots at a concentration of 5 × 10^6^ cells/mL (Figure 6). 

## 4. Discussion

This study aims to elucidate the impact of zinc on yeast cell growth and biofilm formation/development in two strains of *H. capsulatum*, G186A and EH-315. Transition metals such as Fe^2+^, Zn^2+^, Mn^2+^, and Cu^2+^ are essential for most organisms and play structural and catalytic roles in macromolecules involved in several fundamental biological processes. Therefore, an essential aspect of microbial infections is the assimilation of micronutrients in the intracellular environment of the host. In this regard, metal ions such as zinc (Zn^2+^) play a vital role in both the host and the pathogen [10,23,26]. 

The tests were conducted by adding zinc (Zn^2+^) or reducing it using a chelator (TPEN) with a higher binding constant for the metal in the fungal growth medium. TPEN was used to chelate the zinc in a medium and to determine the lowest concentration of the chelator that could inhibit fungal growth. TPEN is a chelator capable of binding to different metal ions and causing cell apoptosis. However, it has a higher affinity constant (K) for heavy metals such as zinc (K = 18.7), followed by iron (K = 14.6), calcium (K = 4.4), and magnesium (K = 1.7) [41]. In addition, it can permeate through the cell membrane and reach the interior of the fungal cell, causing a profound intracellular transient zinc depletion [34,42,43,44]. The cytotoxic effect of TPEN is thought to be due to the chelation of intracellular zinc, which interferes with the function of essential metalloproteins [43]. Laskaris et al. [44] demonstrated that TPEN at 0.125 mg/mL (0.3 µM) effectively reduced the growth of *Aspergillus fumigatus*, causing the atrophy of fungal cells but not completely inhibiting growth. Additionally, the study demonstrated that the chelator binds to zinc ions, as evidenced by the fact that when recovering fungus growth with different metal ions, only zinc is effectively restored when the effect of the chelator is ceased. 

The findings suggest that TPEN at a concentration of 10 µM successfully reduced the growth of *Histoplasma*. This chelator has also been studied in other works, and different concentrations have shown satisfactory results with other fungi [32,33,44,45]. Dade et al. [46] used 10 µM of TPEN to chelate zinc in *Histoplasma* Macrophage Medium (HMM), demonstrating its efficacy in inhibiting *Histoplasma* growth. Another study by Winter et al. [32] found that a higher concentration of TPEN, specifically 100 µM, led to the failure of *Histoplasma* cells to grow and develop. Previous studies have emphasized zinc and iron’s roles in the fungus’s survival within macrophages during host infection [32,46,47,48]. Manipulating the availability of metal ions through chelation is a potential strategy to control fungal growth and infection.

Several authors have described the quantification of metabolic activity using the XTT assay to verify cellular behavior in different situations, such as treatment and growth kinetics. This assay quantifies viable cells in planktonic cultures and biofilms, both eukaryotic and prokaryotic. In this assay, cell viability is calculated as a function of metabolism, which changes the color of the XTT solution from yellow to orange in the presence of viable cells. Despite its popularity, problems regarding intra- and interspecies variability have been reported by authors [49,50]. 

In our work, the XTT assay demonstrated compelling results concerning the metabolic activity of *Histoplasma* strains with different growth conditions. During planktonic growth, the G186A strain showed higher metabolic activity in a zinc medium than the control medium (HAM-F12) and an increase in cell density. The metabolic activity of the EH-315 strain remained similar between the HAM-F12 medium with added zinc and the control condition over 144 h. The cell density of the strain EH-315 increased during the final stages of cultivation (at 120 and 144 h) in the control medium. Both strains displayed a decrease in metabolic activity and cell density when TPEN was introduced to the HAM-F12 medium. Statistical differences in growth between the strains in the medium with TPEN became more pronounced starting from 48 h, as observed in other studies [10,33,34].

Studies on the growth and development of dimorphic fungi such as *H. capsulatum* are limited, especially regarding growth and sessile development (biofilm). Our group was the first to demonstrate the ability of *H. capsulatum* to form a biofilm [12]. In addition, our results showed that *Histoplasma* can infect lung cells and present cell clusters characteristic of biofilms [12]. Another study by our group showed that HAM-F12 favored the development and growth of these sessile communities compared to other media used. The authors investigated the difference between the media and observed the presence of zinc ions in HAM-F12 [13]. 

Based on previous results and literature, this study investigated the effects of zinc (both excess and deficiency) on the initial and mature biofilms of *Histoplasma*, using various assays, including the XTT reduction assay, biomass, quantification (crystal violet), and extracellular matrix quantification (safranin), as well as electron microscopy images (cell morphology) and confocal microscopy. Biomass and extracellular matrix production remained stable in both strains. This result was expected because crystal violet is a dye that binds to the surface of molecules, staining both live and dead cells, polysaccharides, and the extracellular matrix, and safranin stains polysaccharides and the extracellular matrix [13,50,51]. The results highlight the importance of zinc for cell growth while indicating that the overall production of biomass and extracellular matrix remains relatively consistent, as indicated by the staining techniques used in the analysis.

The microscopic analyses complemented the results of the colorimetric assays and provided important additional insights. SEM allows the evaluation of detailed surface morphology at high magnification but involves the degradation of the native structural features due to the fixation and dehydration steps performed during sample preparation. CSLM allows visualization of two- and three-dimensional (3D) architecture and the measurement of biofilms. For both G186A and EH-315 strains, the formation of an initial biofilm and a dense biofilm after 144 h was evident under control conditions and in the presence of zinc in the HAM-F12 medium. With the TPEN chelator, a decrease in the mitochondrial activity and the quantification of strain cells was observed. In addition, small hyphae are visible at 24 h for both G186A and EH-315. After 144 h, the chelator inhibited the formation of a dense and massive biofilm, as seen in the control conditions and with zinc for both strains. It can be assumed that reducing metal ions led to decreased metabolic respiration, leading to no more mycelia formation and only a few yeasts. Differences may occur between strains due to variable sensitivity to temperature and the development of distinct genetic mechanisms during morphological transition [52]. Both strains displayed significant hyphal elongation in mature biofilm conditions, including in control conditions and in a medium with the addition of zinc. *Histoplasma* can exist in both yeast cells and hyphae form, and its morphological transitions constitute a significant aspect of the fungus’s capacity to cause disease through inhaled conidia and mycelial fragments [53,54]. 

The dimorphism of the *Histoplasma* fungus represents its functional differentiation, with the hyphal capable of penetrating and colonizing environments such as soil, enhancing nutrient absorption, and producing conidiophores to form and release conidia {52]. In contrast, the yeast form is smaller and more compatible for housing the phagosomes [55]. Differentiation in yeast represents the programming of the fungus to infect host cells and survive in their interior for a more extended period compared to its natural form in mycelium in the environment [54]. 

Studies that have performed these types of analyses using metal ion chelators have also found variable and similar results for fungal species. Harrison et al. [25] investigated the blocked or triggered transition between yeast and hyphal cell types in *C. albicans* and *C. tropicalis* biofilms under different growth conditions using metal ions. In the presence of Zn^+2^, the morphotype of hyphal cells was triggered in *C. tropicalis*, while biofilms of *C. albicans* exposed to zinc showed more yeast cells. In *C. albicans*, Kurakado et al. [45] examined the effects of adding zinc at concentrations from 1 to 100 µM and TPEN at concentrations from 0.01 to 100 µM on the formation of *C. albicans* biofilms. The authors found an increase in biofilm formation in the presence of zinc and a decrease in biofilm formation with a chelator. Another study by Kurakado et al. [47] showed an effective inhibition of biofilm formation of *Trichosporon asahii* using TPEN in planktonic cells. TPEN inhibited biofilm formation and hyphal elongation in *T. asahii*, and zinc-induced elongation enhanced biofilm formation, suggesting that *T. asahii* uses zinc as an environmental indicator. In our work, the amount of zinc present in HAM-F12 and adding more zinc could induce morphological stress of the *Histoplasma* strains so that hyphal formation occurred. Further studies with metal ions and other micronutrients should be conducted to deepen the studies on the morphology of this microorganism. 

Congo Red (CR) assays for *Histoplasma* have not yet been described in the literature, but it is known that CR is used to study morphological aberrations in filamentous fungi and mainly to evaluate mutant strains treated with antifungal agents or compounds with therapeutic potential [55,56]. Congo Red is a red dye with a high affinity for binding to nascent glucan chains, including β-1,3-glucans in the fungal wall. At higher concentrations (above 0.2 mm), CR binds strongly to chitin chains, preventing them from crystallizing and thus inhibiting their lateral aggregation into microfibrils, which causes growth inhibition. At appropriate concentrations, the binding of CR to cell wall components indicates the growth state of yeast without having a lethal effect [52].

Our studies presented that the clinical strain G186A and wild strain EH-315 have a growth deficit in BHI medium with two concentrations of CR (100 and 150 µM), mainly when originating from the medium with TPEN in planktonic growth. This approach indicates that the chelator induced morphogenic perturbations in the cell walls of *Histoplasma* strains. The cell wall is a dynamic structure that can adapt to different growth conditions and remodel itself during the morphogenetic process, as in the morphological transition of filamentous fungi. Cell wall components of fungal walls, especially chitin and chitosan, effectively chelate environmental divalent metal ions [51]. The adaptive behavior of the cell in response to environmental stress results in changes in the chemical composition and/or structure of the cell wall [57]. Hyphal morphogenesis is closely linked to cell wall synthesis and the polysaccharide composition of the cell wall. Compounds that inhibit or reduce abnormalities in fungal morphogenesis alter cell wall architecture and polysaccharide composition. The primary role of polysaccharides in morphogenesis can be attributed to microfibril polymers. During apical elongation, chitin seems essential to the cross-linking of β-1,3-glucans [58]. More studies need to be carried out to complete the characterization of the influence of metals on the cell wall of *Histoplasma* strains, but the results established that metal chelation can be a powerful strategy for investigating stress mechanisms induced in morphogenesis and the fungal cell wall.

Studies involving metal ion chelators and the Congo Red susceptibility test involve using mutant strains for a gene that captures or absorbs these ions. In a study using strains of *C. parapsilosis*, these yeasts were knocked out for a gene that produces iron reductases. However, wild-type and silenced yeasts were susceptible to an iron chelator (BPS) added to a solid medium. When Congo Red was added to the medium, mutant yeasts showed greater sensitivity to the dye than wild yeasts. The results of this work suggest that the absence of iron reductases causes stress in the yeast fungal wall and that there is a need for metal for this structure [59].

Therefore, the unprecedented observations shown in this study suggest that the presence of zinc ions is essential to stimulate the proliferation of cells of the *Histoplasma* fungus. Furthermore, it was shown that the presence of the chelator TPEN led to a reduction in the metabolic activity of fungal strains, confirming the essentiality of zinc ions for cell growth. Despite limited reports and studies on this fungus, previous research has identified zinc-dependent genes responsible for its growth and development. This study presents new insights into the role of metal ions and biofilm formation in the dimorphic fungus *Histoplasma*.

## 5. Conclusions

Studies conducted on the influence of micronutrients, mainly metallic ions, on the formation of biofilms are recent and have shown remarkable progress. In this work, the biofilm formation of a clinical strain (G186A) and wild strain (EH-315) of H. capsulatum was positively influenced by the addition of zinc to the HAM-F12 culture medium. 

The formed biofilm showed a higher population density, which confers more significant cell cluster formation and may indicate a potentially more virulent biofilm. The depletion of zinc using an intracellular ion chelator (TPEN) promoted a decline in the biofilm formation of both strains, hindering the increase in population density. Future studies are needed to deepen the understanding of the relationship between *H. capsulatum* biofilms, metal ions, and infection. Furthermore, removing zinc hinders fungal biofilm formation and may be a potential therapeutic strategy in conjunction with antifungals to combat histoplasmosis. 

## Figures and Tables

**Figure 1 jof-10-00361-f001:**
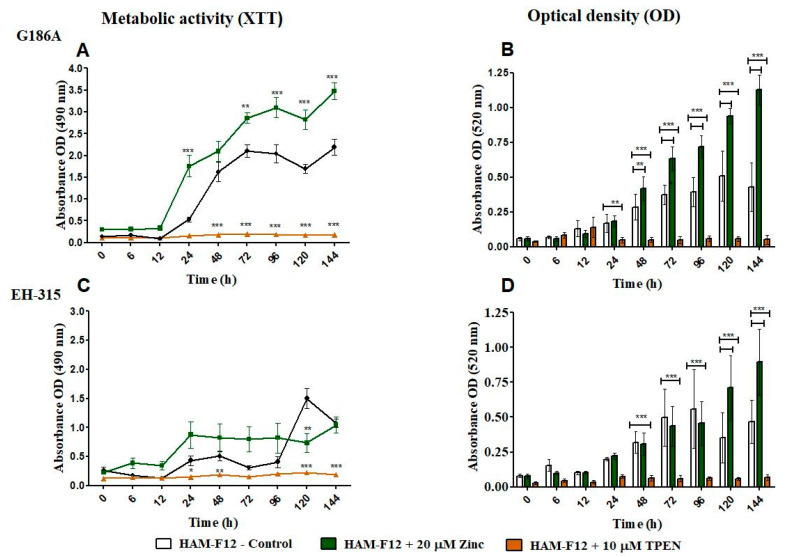
Characterization of planktonic growth of G186A and EH-315 strains of *H. capsulatum* in control medium (control), medium with zinc (Zinc), and medium with TPEN (TPEN). Assay of reduction in metabolic activity by the XTT method for strains G186A (**A**) and EH-315 (**C**). Optical density of cells at 520 nm absorbance for G186A (**B**) and EH-315 (**D**). Error bars indicate a standard experimental condition of adding zinc and TPEN with the control (HAM-F12). (* *p* < 0.05, ** *p* < 0.01, *** *p* < 0.001). Line color black—control (HAM-F12); line color green—HAM-F12 + zinc and line color orange—HAM-F12 + TPEN.

**Figure 2 jof-10-00361-f002:**
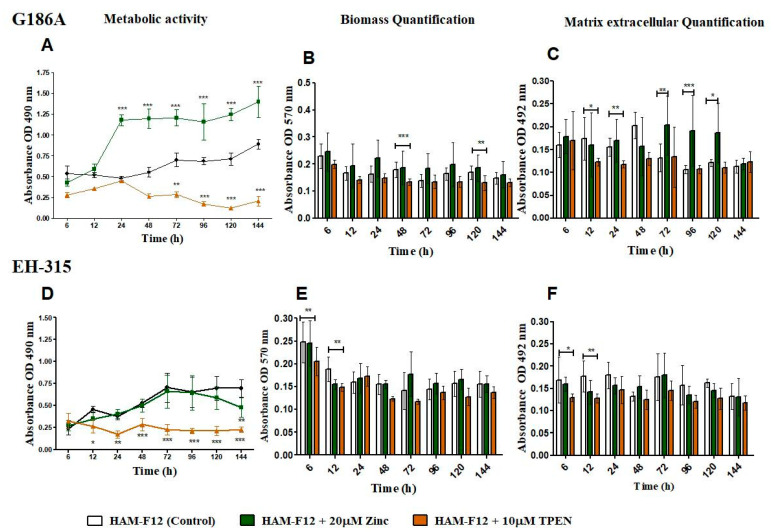
Characterization of biofilm growth of G186A and EH-315 strains of *H. capsulatum* in control medium (control), medium with zinc (Zinc), and medium with TPEN (TPEN). Assay of reduction in metabolic activity by the XTT method for G186A (**A**) and EH-315 (**D**) strains. Quantification of biomass production by crystal violet assay for strains G186A (**B**) and EH-315 (**E**). Extracellular matrix quantification by safranin for G186A (**C**) and EH-315 (**F**). Error bars indicate standard deviations. The *p* values (* *p* < 0.05, ** *p* < 0.01, *** *p* < 0.001) were calculated by comparing the experimental conditions of the addition of zinc and TPEN with the control (HAM-F12). Line color black—control (HAM-F12); line color green—HAM-F12 + zinc and line color orange—HAM-F12 + TPEN.

**Figure 3 jof-10-00361-f003:**
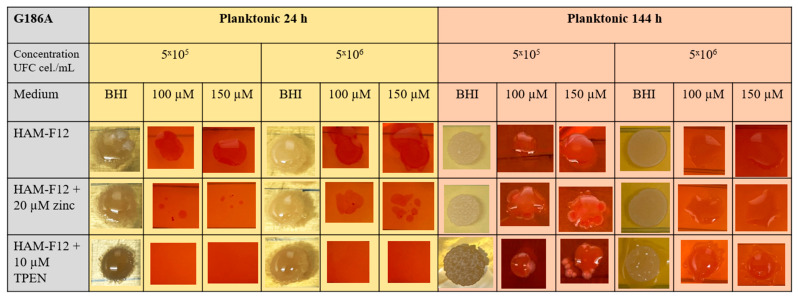
Susceptibility to CR. The assay was carried out at the indicated dye concentrations, and fungal growth was measured by UFC (cell/mL). The images show the G186A strain of *Histoplasma* in planktonic growth at 24 and 144 h.

**Figure 4 jof-10-00361-f004:**
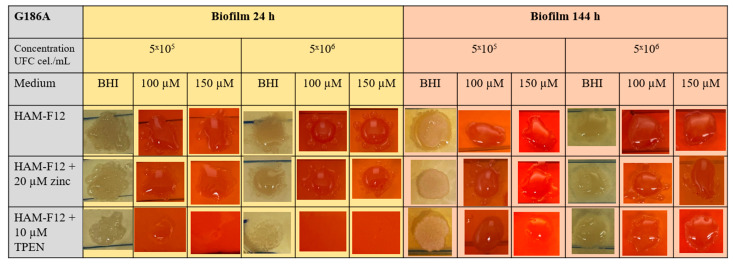
Susceptibility to CR. The assay was carried out at the indicated dye concentrations, and fungal growth was measured by UFC (cell/mL). The images show the G186A strain of *Histoplasma* in biofilm growth at 24 and 144 h.

**Figure 5 jof-10-00361-f005:**
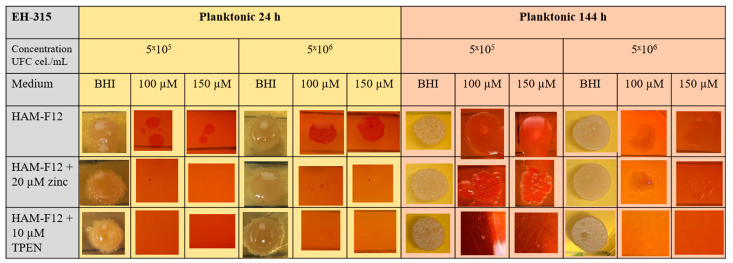
Susceptibility to CR. The assay was carried out at the indicated dye concentrations, and fungal growth was measured by UFC (cell/mL). The images show the EH-315 strain of *Histoplasma* in planktonic growth at 24 and 144 h.

**Figure 6 jof-10-00361-f006:**
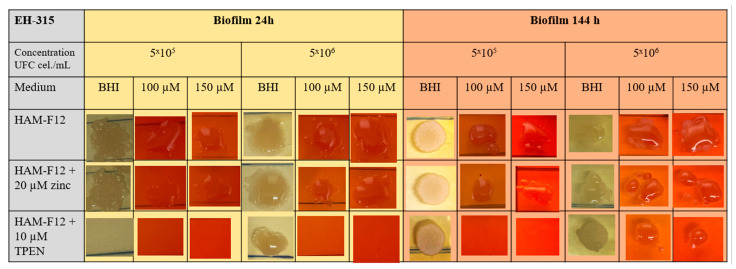
Susceptibility to CR. The assay was carried out at the indicated dye concentrations, and fungal growth was measured by UFC (cell/mL). The images show the EH-315 strain of *Histoplasma* in biofilm growth at 24 and 144 h.

**Table 1 jof-10-00361-t001:** Scanning electron microscopy (SEM) images and confocal microscopy (CLSM) images in two-dimensional and three-dimensional views of strain G186 A of *H. capsulatum* after the formation of the initial biofilm (24 h) (**A**–**I**) and mature biofilm (144 h) (**J**–**R**) in HAM-F12 medium (control), HAM-F12 + 20 µM zinc, and HAM-F12 + 10 µM TPEN. All representative images were magnified 1000× (10 µM). Green arrows—yeast/yellow arrows—hyphae.

G186A			
Time	HAM-F12 Control	HAM-F12 + 20 µM Zinc	HAM-F12 + 10 µM TPEN
24 h SEM	**A** 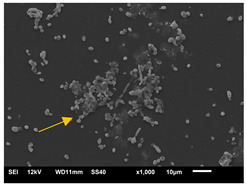	**B** 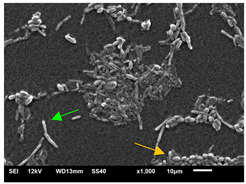	**C** 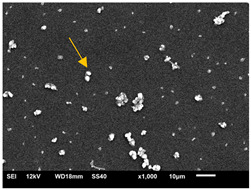
24 h CLSM 2D	**D** 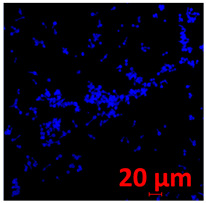	**E** 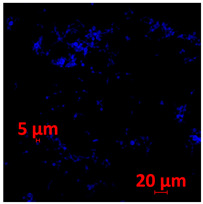	**F** 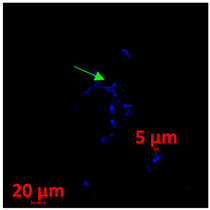
24 h CLSM 2.5D	**G** 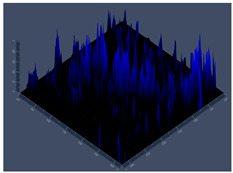	**H** 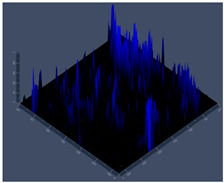	**I** 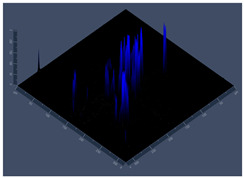
144 h SEM	**J** 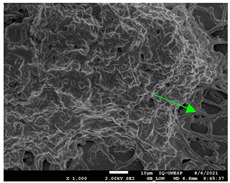	**K** 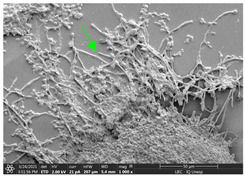	**L** 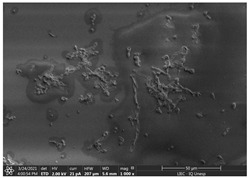
144 h CLSM 2D	**M** 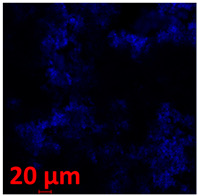	**N** 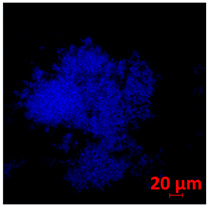	**O** 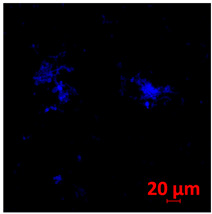
144 h CLSM 2.5 D	**P** 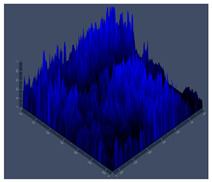	**Q** 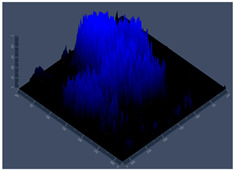	**R** 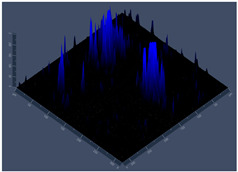

**Table 2 jof-10-00361-t002:** Scanning electron microscopy (SEM) images and confocal microscopy (CLSM) images in two-dimensional and three-dimensional views of strain EH-315 of *H. capsulatum* after the formation of the initial biofilm (24 h) (**A**–**I**) and mature biofilm (144 h) (**J**–**R**) in HAM-F12 medium (control), HAM-F12 + 20 µM zinc, and HAM-F12 + 10 µM TPEN. All representative images were magnified 1000× (10 µM). Green arrows—yeast/yellow arrows—hyphae.

EH-315			
Time	HAM-F12 Control	HAM-F12 + 20 µM Zinc	HAM-F12 + 10 µM TPEN
24 h SEM	**A** 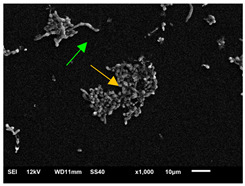	**B** 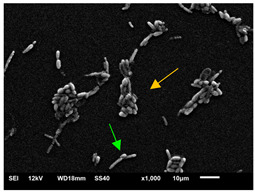	**C** 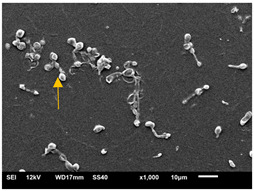
24 h CLSM 2D	**D** 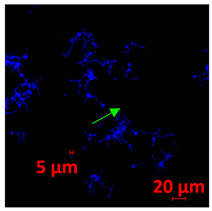	**E** 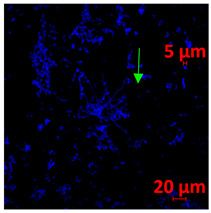	**F** 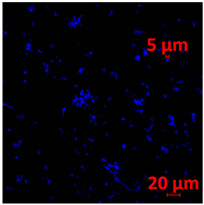
24 h CLSM 2.5D	**G** 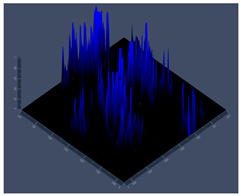	**H** 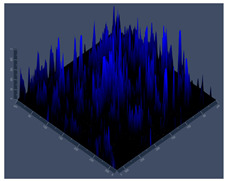	**I** 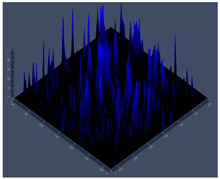
144 h SEM	**J** 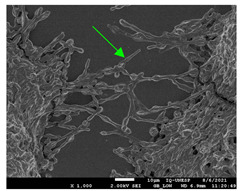	**K** 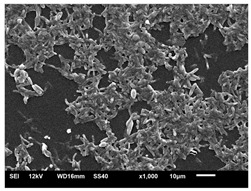	**L** 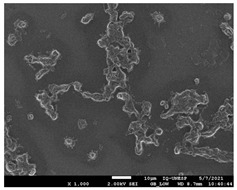
144 h CLSM 2D	**M** 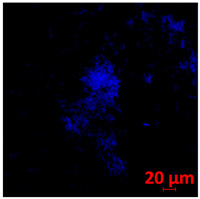	**N** 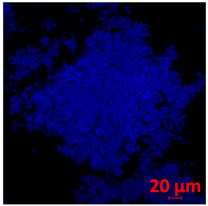	**O** 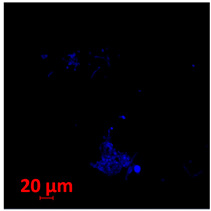
144 h CLSM 2.5 D	**P** 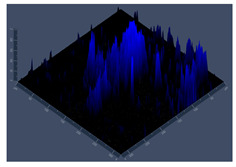	**Q** 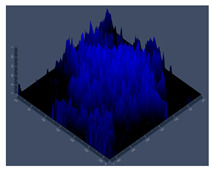	**R** 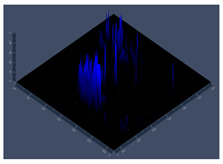

## Data Availability

Not applicable.

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
