# Peer review of "Influence of Zinc on Histoplasma capsulatum Planktonic and Biofilm Cells"

_jof, 2024, doi:10.3390/jof10050361_

Round 1
Reviewer 1 Report (Previous Reviewer 1)
I would like to thank authors for addressing my concern.
I would suggest to put the same color for control in all graphs and mention control in (Figure 1A &C) and (Figure 2A &D.
Need a minor improvement in English language throughout the paper.
Author Response
Revisor 1
I would like to thank authors for addressing my concern.
I would suggest to put the same color for control in all graphs and mention control in (Figure 1A & C) and (Figure 2A & D.
Need a minor improvement in English language throughout the paper.
The authors thank you for the considerations of the article. We made the proposed corrections and improved English. The corrects are highlighted in green. We used the bar in white on both figures.

Reviewer 2 Report (New Reviewer)
It is a well done study, however, important issues should be addressed:
1. Introduction: Too long and details more suitable in discussion or a narrative review. Also, too many cited references (41).
2. Methods: hard to read especially with too many tools and details, I suggest some summary in tables or appendix.
3. Results: Usually the figures title and legend are under the figures..
4. Improve the flow of the discussion.
Needs improvement
Author Response
Revisor 2
It is a well done study, however, important issues should be addressed:
- Introduction: Too long and details more suitable in discussion or a narrative review. Also, too many cited references (41).
The authors are grateful for all corrections and suggestions made. All modifications are highlighted in pink. The introduction has been shortened and the list of references has been revised.
- Methods: hard to read especially with too many tools and details, I suggest some summary in tables or appendix.
We added a Figure 1 and Figure 2 on supplementary material with a flowchart demonstrating from the cultivation of the strains in each condition to the assays performed for the planktonic form and the biofilm growing form.
- Results: Usually the figures title and legend are under the figures.
We corrected the position of titles and subtitles. Thank you.
- Improve the flow of the discussion.
The discussion was revised, and some sentences were removed and re-written to make the manuscript more fluid.

Reviewer 3 Report (New Reviewer)
Dear authors:
I consider this to be a good experimental work, which tries to explain the role of metals such as zinc in the development of Histoplasma and how they contribute to the pathogenic capacity of this fungus and how its depletion with a chelator inhibits the development of biofilm and planktonic cells.
There are only minor redaction mistakes:
On line 219 place 5 x105 and 5 x 106 as superscript
On line 225, it should say p<0.05
Histoplasma should always be written in italics (lines 526, 554)
Correct: should say analysis instead of analyzes (line 512)
Author Response
Reviewer 3
There are only minor redaction mistakes:
On line 219 place 5 x105 and 5 x 106 as superscript – Line 212 now.
On line 225, it should say p<0.05 – line 218 now.
Histoplasma should always be written in italics (lines 526, 554) – corrected now 494 and 521
Correct: should say analysis instead of analyzes (line 512) line 481 now
The authors are grateful for all corrections made. All modifications were made and highlighted in red throughout the manuscript.

Reviewer 4 Report (New Reviewer)
Dear Editor and Authors,
The article on title “Influence of zinc on Histoplasma capsulatum planktonic and biofilm cells” is really interesting and present the influence of zinc and the metal chelator TPEN on the growth of Histoplasma in planktonic and biofilm form. To date, there is a small number of publications regarding this pathogen. Therefore, I believe that the work should be published in a magazine. I find the article is pretty good written and deserves to be published in Journal of Fungi. I recommend major revison which I support with the following arguments:
Abstract (paragraphs):
24: Virulence factors were not determined, so I would not mention biofilm virulence.
· I would highlight the best result/conclusion
· I have a question: how does TPEN affect the human body?
Introduction:
67-75: It looks like a transferred fragment! Different citation pattern and language.
Very decent introduction, but requires emphasizing at the end the importance of the research and its uniqueness.
Materials and Methods
· Data for CLSM imaging need to be supplemented. (λ)
Results:
· Recommends performing an additional BOAT test in the presence of TPEN.
· Results from SEM and LSM should be apart. Imaging for adhesion at 2.5 D for the first hours (24H) does not make sense because of the piling effect (signal fracturing effect). For 144H, definitely yes. Consider adding an intermediate time.
· Consider changing the imaging color (auto color in software option) or brightening the photos. (The details of the structures in CLSM are hardly visible).
Discussion:
I suggest emphasizing the originality of the results obtained.
I recommends including the following articles:
https://www.nature.com/articles/s41598-021-88244-1
https://link.springer.com/article/10.1007%2Fs00253-020-10349-w
Author Response
Reviewer 4
Abstract (paragraphs):
24: Virulence factors were not determined, so I would not mention biofilm virulence.
- I would highlight the best result/conclusion
- I have a question: how does TPEN affect the human body?
We appreciate all corrections and suggestions. All changes made to the text highlighted in blue. The sentence in line 24 of the Abstract was removed. And the result and conclusion was improved.
Regarding the action of TPEN in the body, few studies have demonstrated that the cheator in certain concentrations can be toxic to alveolar macrophages (WINTERS, M. S., et al, 2010 - doi: 10.1086/656191), however, the authors themselves suggest that more concentrations and studies must be carried out to verify this hypothesis. In our group, there is an investigation into the effects of chelator in cell cultures (in vitro), in addition, it aims to expand to alternative animals (in vivo).
Introduction:
67-75: It looks like a transferred fragment! Different citation pattern and language.
Very decent introduction but requires emphasizing at the end the importance of the research and its uniqueness.
We corrected and put the text for line 55 until 62.
“Biofilm formation occurs in a sequence of processes that will overlap during its development. Despite the use of sequenced and labeled stages of biofilm formation and development, the actual processes that occur under native conditions are much more complex, dynamic, and diverse (Guzman-Soto, et al, 2021). Many authors have demonstrated that the addition of metal ions or the removal of these ions disrupts the formation of their respective biofilm [25-27]. Studies have shown that metal ions alter the structures of Candida tropicalis and C. albicans biofilms and increase the transition between yeast form and hypha cells. This increases invasiveness and alters financial virulence [51]”.
Materials and Methods
Data for CLSM imaging need to be supplemented. (λ)
The information was cited in lines 210-213.
“After the staining time, the wells were washed with PBS, and the samples were analyzed at a wavelength of 380 nm using a Confocal Fluorescence Microscope (Carl Zeiss LSM 800) at the Faculty of Dentistry of UNESP, Araraquara, SP, Brazil. and analyzes were performed using Zen Blue 3.2 software (Carl Zeiss, Jena, Germany).”
Results:
Recommends performing an additional BOAT test in the presence of TPEN.
Our current study is introductory in research on metal ions and the formation of biofilms or dimorphic fungi. We considered the suggestion of a BOAT test using TPEN, in addition, more studies with different ions, chelators and cultivation conditions are being carried out by the group.
Results from SEM and LSM should be apart. Imaging for adhesion at 2.5 D for the first hours (24H) does not make sense because of the piling effect (signal fracturing effect). For 144H, definitely yes. Consider adding an intermediate time.
Answer: We added the 2.5 D image at 24 h, to make it easier for the reader to visualize the behavior of cells in an initial biofim. And precisely, these two times were chosen, since previous studies (Pitangui et al, 2012 doi:10.3389/fmicb.2020.01455; Fregonezi et al, 2020 doi:10.3389/fcimb.2020.591950 and Gonçalves et al, 2020 doi:10.1080/08927014.2012.703659), and this study, demonstrated that within 24 hours Histoplasma had the beginning of a biofilm (initial biofilm) and the time of 144 h was the maturation of these communities. Unfortunately, at this time it will not be possible to add new analysis times due to problems with the university's equipment, but we will consider this suggestion for future work in addition to new conditions.
Consider changing the imaging color (auto color in software option) or brightening the photos. (The details of the structures in CLSM are hardly visible).
Answer: The authors thank you for your suggestions and we use the software to improve the color of images.
Discussion:
I suggest emphasizing the originality of the results obtained.
I recommends including the following articles:
https://www.nature.com/articles/s41598-021-88244-1
https://link.springer.com/article/10.1007%2Fs00253-020-10349-w
We appreciate suggestions regarding the discussion and improve the discussion.

Reviewer 5 Report (New Reviewer)
The authors investigated the effect of zinc and the metal chelator TPEN on the growth of Histoplasma in planktonic and biofilm form. Overall, the article is interesting and of high scientific potential, the results show that zinc increases metabolic activity, cell density, and cell viability.
I recommend to accept the manuscript after correction of the following remarks:
Give the full names and the abbreviation in brackets
Line 66 and 70 You did not include 2 authors
Line 87-88 Paraphrases the sentence
Line 96 pyridozine?
The subsection is italicized and aligned (use template)
Extensive editing of English language required
Author Response
Reviewer 5
Comments and Suggestions for Authors
The authors investigated the effect of zinc and the metal chelator TPEN on the growth of Histoplasma in planktonic and biofilm form. Overall, the article is interesting and of high scientific potential, the results show that zinc increases metabolic activity, cell density, and cell viability.
I recommend to accept the manuscript after correction of the following remarks:
Give the full names and the abbreviation in brackets
Answer: The changes are highlighted in yellow.
Line 66 and 70 You did not include 2 authors
Line 87-88 Paraphrases the sentence
Answer: We rewrote the sentence and included the missing authors in the text. From line 62 to line 74.
The subsection is italicized and aligned (use template)
Answer: We corrected the subsection. Highlighted in yellow.
Line 96 pyridozine?
Answer: Line 83 now – pyridoxine
Comments and Suggestions for Authors: Extensive editing of English language required
We appreciate suggestions and made the corrections.

Round 2
Reviewer 2 Report (New Reviewer)
Comments addressed and manuscript improved
Reviewer 4 Report (New Reviewer)
The manuscript has improved, although not all of the reviewer's suggestions have been implemented.
Reviewer 5 Report (New Reviewer)
The corrections are ok.
Accept in present form
This manuscript is a resubmission of an earlier submission. The following is a list of the peer review reports and author responses from that submission.
Round 1
Reviewer 1 Report
The design of the experiments need improvement.
1. I suggest including positive control to show the significance of TPEN.
2. I am concerned why the percentage of viable cells (control) is very low whereas the strain cultured with only complete medium, the experiment needs to explain more clearly.
3. The differences of viable cell percentage are not clear after TPEN treatment, authors showed a significance difference but, in the figure, it seems like very less.
4. In the figure 1 the Metabolic Activity does not contain any unit and not clear.
5. In Figure 1 the optical density should be replaced by the cell density to make it clearer.
6. Figures should contain different sub figures and they should be named as A(i) or convenient numbers.
7. The reduction of biomass in Figure 2 is not consistence after TPEN treatment and the effect on Extracellular matrix is also not convincing.
Reviewer 2 Report
The study presented the influence of zinc cations on planktonic growth and biofilm structure of the Histoplasma capsulatum fungus.
1/ The information contained in the Introduction is very vague, particularly regarding the role of zinc in the metabolism of the analyzed microorganism. The Introduction should be significantly deepened in this regard. (See example work: Assunção LDP, Moraes D, Soares LW, Silva-Bailão MG, de Siqueira JG, Baeza LC, Báo SN, Soares CMA, Bailão AM. Insights Into Histoplasma capsulatum Behavior on Zinc Deprivation. Front Cell Infect Microbiol. 2020 Nov 30;10:573097. doi: 10.3389/fcimb.2020.573097).
In the Result section:
2/The concentration of 10 uM chelator led to the disappearance of cell viability, so what was studied in this regard at this concentration? The discussion also did not refer to other chelators used in the literature that do not cause total cell inactivation.
3/ It is not clear what the term cell viability means, whether it is the measurement using trypan, as in the methodological description, or staining in which resazurin was used. Usually, the term refers to the use of the classic cell viability assay – XTT (https://doi.org/10.1016/j.bioflm.2022.100090), which, however, the authors applied to determine mitochondrial activity, taking advantage of the fact that the assay is based on the activity of mitochondrial dehydrogenases.
4/In contrast, to determine mitochondrial functionality, other assays would have to be used, such as mitochondrial membrane potential, oxygen consumption or intracellular Ca2+ content (see - https://doi.org/10.3892/ijmm.2022.5182).
5/ In the biofilm figure (figure 2), the scatter is so large that it is difficult to infer the statistical significance indicated for single time intervals. It would be necessary to verify the results of the statistical analysis.
6/ For SEM analysis of G186 A of H. capsulatum (Fig. 3) no influence of zinc on the biofilm composition is detected.
7/ The results in Fig.4 are surprising, as for initial biofilm increased filamentation in the presence of zinc is observed and for mature biofilm filamentation is reduced but the authors don't explain it.
8/ Confocal microscopy images of strains EH-315 for mature biofilm (Fig.8) do not appear to correspond to the results from SEM
9/ the SEM results do not correspond with the results determining the formation of the biomass and cell matrix, as well as with the conclusion regarding the influence of TPEN.
There is much information in publications on fungal cell functioning in the depletion of zinc content, but in the discussion part no comparisons with those data are presented. Moreover, the data for biofilms do not correspond to each other. In the discussion part, no analysis is performed concerning differences in the shape and appearance of cells within the biofilm when exposed to excess zinc and in the presence of chelator. The discussion must be reconstructed.